# Identification Method of River Blocking by Debris Flow in the Middle Reaches of the Dadu River, Southwest of China

**Zhi Song** [1,2,3]**, Gang Fan** [4]**, Yanni Chen** [2] **and Darui Liu** [4,*]

[1] College of Earth Science and Engineering, Shandong University of Science and Technology, Qingdao 266590, China; songzhi520@163.com
[2] School of Tourism and Urban-Rural Planning, Chengdu University of Technology, Chengdu 610081, China; chenyanni@stu.cdut.edu.cn
[3] Chengdu Center of China Geological Survey, Chengdu 610081, China
[4] College of Water Resource and Hydropower, Sichuan University, Chengdu 610065, China; fangang@scu.edu.cn
[*] Correspondence: liudarui@stu.scu.edu.cn

**Abstract:** Debris flow is a typical natural disaster in the middle reaches of the Dadu River, which seriously threatens the safety of life and property of local residents. However, there is currently a lack of a comprehensive analysis methods applicable to the blockage of river channels by debris flow in the Dadu River basin, limiting disaster prevention and mitigation in this area. Based on previous large-scale model tests carried out in the middle reaches of the Dadu River, the debris flows are divided into dam-type debris flows and submerged debris flows. The calculation formulas for the maximum travel distance of the two kinds of debris flows entering the river are obtained via theoretical derivation. The formulas for calculating the length and volume of debris flow accumulation are derived, and the relationship between the debris flow loss coefficient and river blocking degree in the middle part of the Dadu River is analyzed. An identification method of river blocking by debris flow is put forward in this study. By calculating the maximum blocking degree, S (the ratio of the maximum driving distance of the debris flow to the width of the river), and the volume of the source materials needed to form a debris flow dam under the conditions that the debris flow does not reach the opposite bank ($V_1$), reaches the opposite bank but does not block the river ($V_2$), and reaches the opposite bank ($V_3$), the form of debris flow blocking the river is distinguished. When $S = 1$, $V > V_3$, complete blockage occurs; when $S = 1$, $V > V_2$, the river is mostly blocked; when $S < 1$, $V > V_1$, the river is half-blocked. This study established an identification method of river blocking by debris flow, providing a basis for early warning for river blocking and disaster prevention in the middle reaches of the Dadu River.

**Keywords:** river blocking; identification method; debris flow; Dadu River





## 1. Introduction

Debris flow is a special type of torrent that carries a large amount of sediment, rocks, and other materials, and that is caused by short-term heavy precipitation in mountainous areas and snow-melting [1–4]. Furthermore, after debris flows enter rivers, they can trigger secondary disasters, such as dammed lakes, which can threaten hydropower stations, transportation infrastructure, and the lives and properties of local residents. Debris flow disasters have caused enormous losses worldwide. In May 1970, a debris flow disaster occurred in the town of Yange, Peru. The flow speed was nearly 100 m/s, and $5000 \times 10^4$ m$^3$ of solid material was washed away, resulting in nearly 20,000 deaths [5]. On 15 December 1999, heavy rain hit all parts of Venezuela, causing dozens of gullies to develop large-scale debris flows at the same time, and resulting in 30,000 casualties and enormous economic losses of up to 10 billion US dollars [6]. On 19 August 2019, Wenchuan County in the Sichuan Province of China suffered multiple debris flow disasters caused by heavy precipitation; approximately 446,000 people were affected, and this event caused nearly 15.89 billion yuan of direct economic losses [7,8].

The Dadu River is located in northwestern Sichuan Province. The middle reaches of the Dadu River have deep river valleys, strong tectonic activity, and complex stratigraphic lithology, resulting in the frequent occurrence of debris flow disasters in this region. On 23 July 2009, a debris flow occurred in Xiangshui gully, Kangding city, and completely blocked the Dadu River, causing 16 deaths, 38 missing, and considerable property losses [9]. On 12 July 2012, a debris flow occurred in Tangjia gully, Shimian County, Yaan city, and blocked the Dadu River, resulting in two deaths and five missing persons [10]. On 4 July 2013, the Xiongjia gully debris flow completely blocked the Zhuma River and caused 19 deaths [11].

Hence, identifying the risk of river blocking by debris flows is important for disaster prevention and mitigation in the middle reaches of the Dadu River. At present, there are four main methods for studying the mechanism of river blocking by debris flows: (1) experimental studies using flume tests [12–16]; (2) statistical models based on field investigations [17–19]; (3) numerical simulations; and (4) theoretical analysis [20–22]. However, there are few practical studies on the identification of river blocking by debris flows in the Dadu River basin, which serves as one of the major hydropower bases in China. Therefore, a more applicable and comprehensive identification method is needed for the Dadu River to reduce the loss of life and property caused by debris flow disasters in the basin.

In this study, the interaction mechanism between the debris flow and the water flow after the debris flow enters the river is analyzed based on the field investigation in the middle reaches of the Dadu River, and the debris flows occurred in this area are divided into a barrier type and a submerged type according to whether the debris flow into the river is below the water surface and the characteristics of the debris flow. This study takes the three-dimensional spatial relationship between tributaries and main streams into account, formulas for calculating the travel distance of different types of debris flows are established, and finally, a comprehensive identification method for debris flow interception under different models is proposed.

## 2. Study Area

The study area is the middle reaches of the Dadu River (Kangding–Shimian section), covering an area of 6183.33 km$^2$. The study area is crisscrossed by water systems and numerous tributaries, such as the Moxi River, Wasigou, Nanya River, Anshun River, and Tianwan Gou. The valleys are deep, the maximum elevation difference is 5200 m, as illustrated in Figure 1. The middle reaches of the Dadu River are located in the transition zone from the Sichuan Basin to the Qinghai–Tibet Plateau. The middle reaches of the Dadu River are divided into three sections, i.e., the Kangding section, the Luding section, and the Shimian section. The middle reaches of the Dadu River are located in the transition zone from the Sichuan Basin to the Qinghai–Tibet Plateau. Affected by the southeast and southwest monsoon and the cold air of Qinghai–Tibet Plateau, the vertical climate difference is obvious. The areas below 1800 m altitude in the study area have tropical monsoon climate and experience local heavy rainfall. There are six tributaries in this area, and each tributary may have different confluence characteristics. The average annual total flow in this section is 330.56 × 10$^8$ m$^3$/s, the average annual flow is 1218 m$^3$/s, and the maximum peak flow is 6050 m$^3$/s. The location of the study area is shown in Figure 1.

The middle section of the main stream of the Dadu River flows along the Dadu River fault zone, which is a zone of abrupt changes in topography and landforms that constitutes the eastern boundary of the Qinghai–Tibet Plateau. The middle reaches of the Dadu River are located on the eastern margin of the Qinghai–Tibet Plateau, and strata other than Cretaceous and Cambrian strata are exposed. Macroscopically, the middle reaches of the Dadu River are located at a tectonic intersection with complex structures. The north side is the Sichuan–Qinghai block, the west side is the Sichuan–Yunnan block, and the east side is the South China block. The NW-trending Xianshuihe fault, the NE-trending Longmenshan fault and the N–S-trending Anninghe fault constitute a Y-shaped structure in the Shimian section.

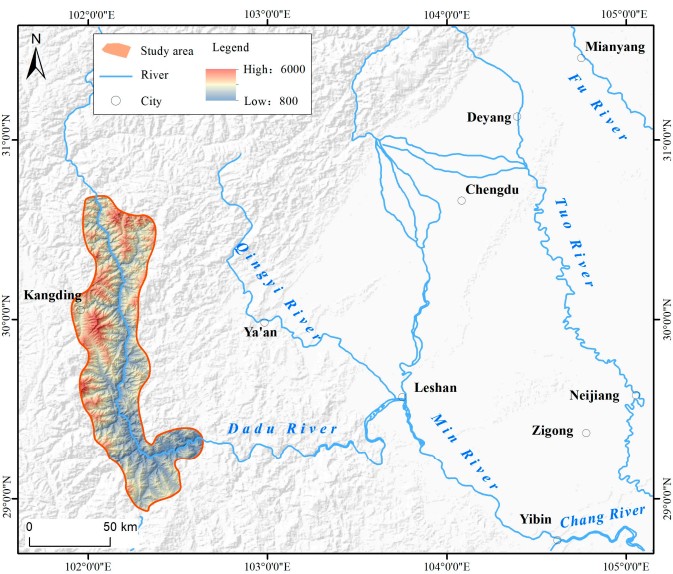

**Figure 1.** Location of the study area.

Field investigation shows that there are a total of 441 debris flows in the study area, and these debris flows can be divided into four categories based on the area of the debris flow deformation region, i.e., extra-large, large, medium and small (Figure 2).

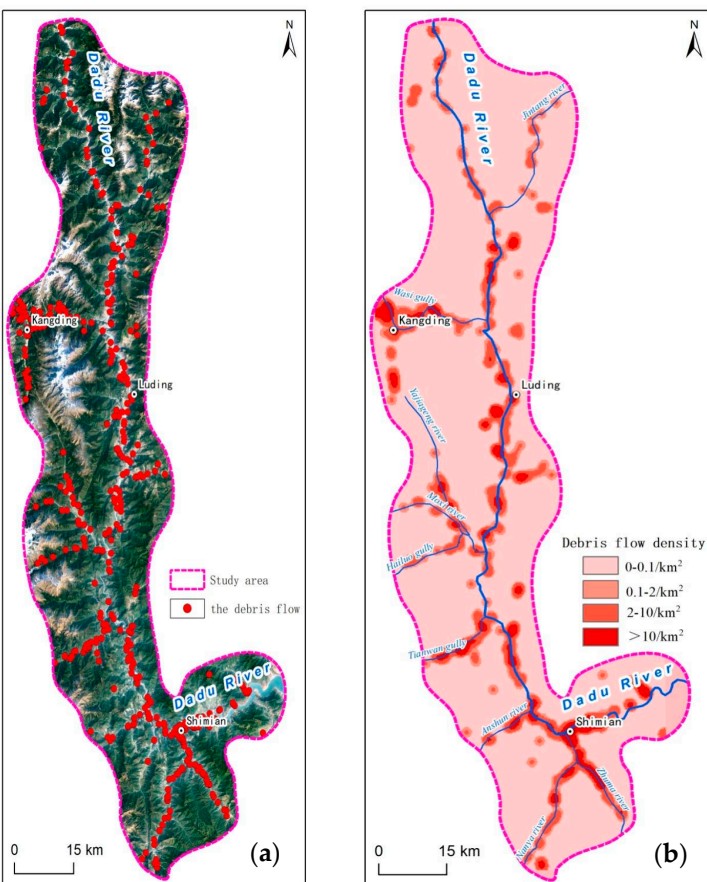

**Figure 2.** Illustration of the study area: (**a**) distribution of the debris flows; and (**b**) debris flow density in the study area.

### 3. Type of Interaction between the Debris Flow in a Branch Channel and the Main River

Debris flows tend to accumulate on flatter terrain, and this is caused by two factors. First, when the debris flows rush out from a gully, the gravitational potential energy is reduced because the topographic slope decreases; second, the debris flow usually spreads in a fan shape after it stops, leading to an increase in friction. As the middle reaches of the Dadu River have flat topography, debris flows are usually deposited in the river channel. After a debris flow enters the main river, some source materials may not flow downstream together with the river since the debris flow deposits too quickly or the river does not have sufficient capacity to scour and transport the material [23,24]. These source materials can resist river erosion and be deposited in the river channel, thus gradually forming a natural dam and eventually leading to river blockage. This is defined as a dam-type debris flow in this study, as illustrated in Figure 3a. In addition, if the water level of the main river is high, making it impossible for a debris flow entering the main river to form a dam-type debris flow, the debris flow will rush into the main river and be submerged, forming a submerged-type debris flow, as illustrated in Figure 3b.

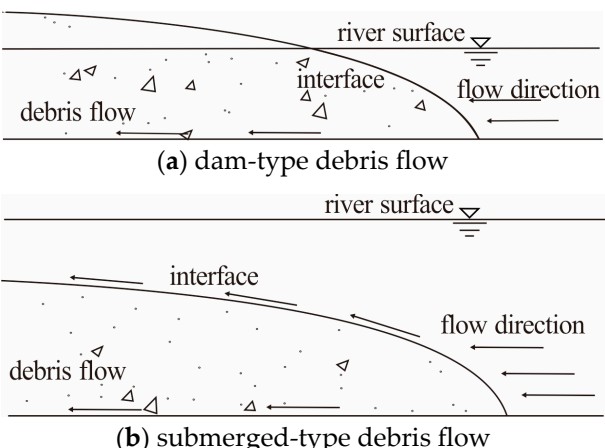

(**a**) dam-type debris flow

(**b**) submerged-type debris flow

**Figure 3.** Two typical motion types of debris flows in the middle reaches of the Dadu River after entering into the main river channel.

The flow field of the debris flow is disordered, and the intermixing effect is difficult to describe quantitatively. To facilitate its theoretical description, the mixing effect at the interface between debris flow and water flow is often ignored. Instead, it is considered a Newtonian interaction with different densities. Considering the different viscosities of the debris flow, it is difficult for turbulent flow at the interface between water and the debris flow to disrupt the fluid structure of the high-viscosity debris flows because debris flows with higher viscosity have higher structural force. However, debris flows with higher structural forces can be easily impacted by water flow due to their poor structure, resulting in more intense mixing. In practice, compared with viscous debris flows, dilute debris flows are less likely to block rivers due to their susceptibility to scouring [25]. Therefore, only cohesive debris flows are considered in this study.

For viscous debris flows, there is an interface between the debris flow and the water flow in the river, as the material exchange between them is ignored (Figure 3). When the debris flow rushes into the river, it is subjected to a bypass resistance at the debris flow head and a frictional resistance from the riverbed at the bottom surface of the debris flow. When the debris flow enters the river in a submerged manner, its top surface is subject to gravitational and shear forces from the river flow, the specific direction of which varies depending on the direction of their mutual movement.

## 4. Calculation Model for Dam-Type Debris Flow

### 4.1. Equations of the Motion of a Debris Flow in a Variable-Slope Channel

Takahashi derived the equation for the motion of debris flows in variable-slope channels based on the conservation of momentum theorem [26].

$$
\frac{d}{dt}\left[\frac{1}{2}\left(h+h_f\right)y\frac{\gamma_d}{g}v_d\right] = \frac{1}{2}\left(h+h_f\right)y\gamma_d\sin\theta + \frac{\gamma_d}{g}q_Tv_u\cos(\theta_u-\theta) \\
+ \frac{1}{2}gh_u^2\cos\theta_u\cos(\theta_u-\theta)\left[(\sigma-p)c_{du}K_a+\rho\right] - F \tag{1}
$$

where $t$ refers to the time; g represents the gravitational acceleration; $h$ and $h_f$ stand for the simplified shape parameters for debris flow head in y–z two-dimensional plane (Figure 4); $\gamma_d$ is the gravity of debris flow; $\rho_\sigma$ refers to the density of sand; $\rho$ means the density of water; $c_{du}$ is the volume specific concentration of the debris flow; $F$ is the ground friction force on the debris flow; and $K_a = \tan^2(45° - \phi/2)$ ($\phi$ is the angle of dynamic friction of the debris flow materials) is the coefficient of the active earth pressure.

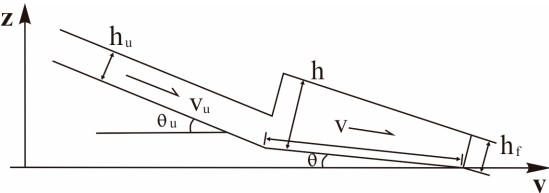

**Figure 4.** Diagram of the motion of the debris flow.

Without considering the change in direction due to the x-directional flow pressure (Figure 5), the equation can be used to calculate the y-directional movement distance of the debris flow in the channel with a variable slope.

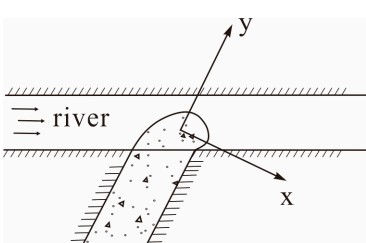

**Figure 5.** Diagram of the debris flow in plane in a channel with a variable slope.

After the debris flow enters the river, its head and body are resisted by both the riverbed and the river flow. Without considering the material exchange between the river flow and the debris flow, the water pressure at the confluence is simplified to hydrostatic pressure, a hydrostatic pressure of the river is added to Equation (1), and the kinetic equation of the motion of debris flow can be obtained, as follows:

$$
\frac{d}{dt}\left[\frac{1}{2}\left(h+h_f\right)y\frac{\gamma_d}{g}v_d\right] = \frac{1}{2}\left(h+h_f\right)y\gamma_d\sin\theta + \frac{\gamma_d}{g}q_Tv_u\cos(\theta_u-\theta) \\
+ \frac{1}{2}gh_u^2\cos\theta_u\cos(\theta_u-\theta)\left[(\sigma-p)c_{du}k_a+\rho\right] - F - f \tag{2}
$$

where $f$ is the resistance of the river flow to the debris flow, which is determined as follows:

$$
f = C_D h_u \frac{\rho v_u^2}{8}\cos\theta + \frac{1}{2}gh_u^2\rho\cos\theta \tag{3}
$$

where the first term on the right side is the drag force of the debris flow when moving in the water, and the second term on the right side is the hydrostatic pressure. Among them, the value of the drag coefficient, $C_D$, is related to the shape of the debris flow and the Reynolds number of the main river flow, which is generally in the range of 0.5–2.5 in

engineering practices. Liu considered the shape of the debris flow head and recommended the values of the drag force coefficient, as listed in Table 1 [27].

**Table 1.** Values for the drag coefficients $C_D$.

| Conditions of the Debris Flow Entering the River | $C_D$ |
|---|---|
| Gentle riverbed, low velocity and low turbulence | 0.5–1.0 |
| Sloping riverbed, medium velocity and medium turbulence | 1.0–1.5 |
| Steeply sloping areas, high velocity and turbulence | 1.5–2.5 |

From the law of conservation of mass, it follows that

$$\frac{d}{dt}\left[\frac{1}{2}\left(h + h_f\right)y\right] = q_T$$

Integrating this and combining the initial value condition $x(t = 0) = 0$, Equation (2) gives

$$\frac{dv}{dt} = -\frac{v}{t} + \frac{Q}{t} - R \tag{4}$$

where

$$R = \frac{(\gamma_d - \gamma_w)g c_{du} n \cos\theta}{(\gamma_d - \gamma_w)c_{du} + \gamma_w}$$

$$Q = v_u \cos(\theta_u - \theta) + \frac{\cos\theta_u \cos(\theta_u - \theta)[(\gamma_d - \gamma_w)c_{du}k_a + \gamma_w]gh_u}{2[(\gamma_d - \gamma_w)c_{du} + \gamma_w]v_u}$$

$$- \frac{\gamma_w \cos\theta g h_u}{2[(\gamma_d - \gamma_w)c_{du} + \gamma_w]v_u} - \frac{C_D \gamma_w v_u \cos\theta}{8[(\gamma_d - \gamma_w)c_{du} + \gamma_w]}$$

Integrating Equations (1)–(4) from 0 to $t$ and substituting the boundary conditions yields $v(t = 0) = 0$, giving:

$$y = -\frac{1}{4}Rt^2 + Qt \tag{5}$$

Substitute yields the formula for the maximum transport distance, as follows:

$$L = \frac{Q^2}{R} \tag{6}$$

*4.2. Analysis of Debris Flow Movements Considering the Spatial Relationship of the Channel*

4.2.1. Maximum Travel Distance at the Cross-Section of the Channel

Equation (6) is a method for calculating the travel distance of a debris flow in a channel with variable slope in a two-dimensional plane, which is applicable when the debris flow intersects the river at a right angle. However, debris flows often rush into rivers at a certain angle. Therefore, the movement calculation model is simplified to give a reasonable travel distance calculation for different spatial relationships.

As shown in Figure 6, after the debris flow enters the river, the movement track of the debris flow head is actually curved due to the fluid and bed forces. The complex changes in the head shape during curved movement will have a significant impact on both the resistance and dynamics equations. To simplify the process, the x- and y-directions are considered to be separate and without interaction, and the calculation of the y-direction travel distance still uses Equation (6), where the angle of the channel is calculated based on the spatial relationship in the plane, resulting in

$$\cos\theta = \sqrt{\frac{\tan^2\beta + 1}{\tan^2\beta + \sin^2\alpha + 1}} \tag{7}$$

where $\alpha$ is the longitudinal slope angle of the river, and $\beta$ is the angle between the debris flow and the river channel.

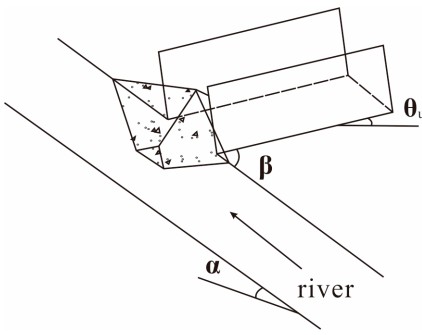

**Figure 6.** Illustration of the spatial relationship between the debris flow and the river.

The equation of the motion for the x-directional debris flow is

$$\frac{1}{2}\rho_0 x h_f a = \frac{1}{2}g h_f^2 \rho_0 \cos\theta_x + \frac{1}{2}c_{du}h_f x g(\rho_\sigma - \rho)\sin\theta_x - f_2 \tag{8}$$

where $\rho_0$ is the average density of the debris flow, $\theta_x$ is the slope angle in the x-direction of the river channel, and $f_2$ is the internal resistance of the debris flow.

The angle of slope in the x-direction of the river channel can be calculated with the following equations:

$$\sin\theta_x = \frac{\tan\beta\sin\alpha}{\sqrt{1 + \tan^2\beta(1 + \sin^2\alpha)}} \tag{9}$$

$$\cos\theta_x = \sqrt{\frac{1 + \tan^2\beta}{1 + \tan^2\beta(1 + \sin^2\alpha)}} \tag{10}$$

The composition of debris flows is complex. Therefore, to establish a simple and practical resistance model, this study adopts structural two-phase resistance models to consider the resistance term in debris flows and mainly considers the frictional stress of solid-phase particles, which are independent of shear deformation, to obtain the velocity component in the river width direction as follows:

$$v_l = \left(-\frac{1}{2}Rt + Q\right)\sin\beta + at\cos(\pi - \beta) \tag{11}$$

The farthest distance is reached when the speed is 0, and the time is

$$t_l = \frac{2Q\sin\beta}{R\sin\beta - a\cos(\pi - \beta)} \tag{12}$$

Integrating over $v_l$ gives a maximum distance of

$$l_{\max} = \left(-\frac{1}{4}Rt_l^2 + Qt_l\right)\sin\beta + \frac{1}{2}at_l^2\cos(\pi - \beta) \tag{13}$$

### 4.2.2. Deposit Morphology of Debris Flows

In Figure 7, the length of the debris flow in the x-direction is still unknown, but it can be seen that the shape and size in the x-direction can be calculated by determining the slope values of the front slope and the back slope, and finally the three-dimensional accumulation dimensions of the debris flow can be obtained.

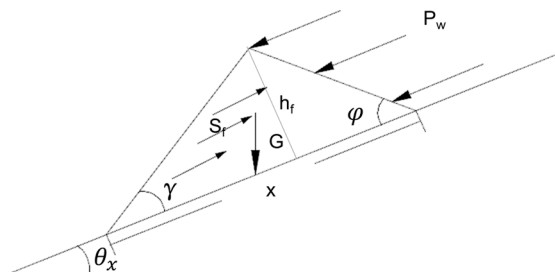

**Figure 7.** Diagram of the x-directional forces.

As the debris flow enters the river, if the slope of the deposit surface is large, the surface flow can erode and further reshape the deposit slope, until forming a stable slope with a critical slope. Kuang derived the calculation equation of the critical slope values $\gamma$ as follows [28]:

$$\tan \gamma = \frac{C(\rho_\sigma - \rho)\tan\phi}{C(\rho_\sigma - \rho) + \rho\left[1 + 0.52\left(\dfrac{q_{JSW}^2}{gd_m^3}\right)^{1/3}\right]} \tag{14}$$

where $C$ is the volumetric concentration of debris flow deposits after the debris flows stop moving, $d_m$ is the mean particle size of the debris flow materials and $q_{JSW}$ is the single width flow rate of the surface flow over the deposit.

## 5. Calculation Model for Submerged-Type Debris Flow

In the analysis of the dam-type debris flow, the forces in the plane of the movement direction are mainly taken into account when calculating the transport distance and determining the degree of river blocking, but the upper water pressure on the debris flow is not considered, so it is not suitable for describing the submerged-type debris flow. Therefore, the dynamic movement characteristics of submerged-type debris flows will be analyzed in this section.

### 5.1. Underwater Movement Characteristics of Debris Flows

Underwater cohesive debris flows have an interface at their top during movement and deposition. Walker divided the deposit profile of underwater debris flows into four gradient processes [29]. (1) The uppermost part is a turbulent layer, in which deposit particles are disorganized and where multiple transport mechanisms (rolling, suspension, dragging, etc.) coexist. (2) When the debris flow travels a certain distance, the deposit begins to show a reverse progression. At this point, the mixing effect of the water reduces the viscosity of the upper layer of the debris flow body and concentrates the coarse particles toward the middle. (3) In the third state, water flow mixing continues to strengthen, and the turbulent flow ripples through the entire debris flow body. The larger particles settle, and the smaller particles cease to move as the flow velocity decreases, thus creating a positive progressive structure. (4) As the debris flow continues to travel and deposit, the deposit profile regressive layers become stable, and the traction of the upper part of the debris flow body increases and gradually becomes the dominant movement.

After entering the main river, the debris flow moves to the opposite bank, and the debris flow body deposits to form a submerged dam, which affects the flow movement, so whether the debris flow body can travel to the opposite bank is an important factor affecting this kind of river blocking mode. However, the underwater motion process of debris flows is complicated, and the external force forms and internal deposition modes are different at different stages. To facilitate theoretical analysis, it is necessary to simplify the process.

*5.2. Movement Model of the Debris Flow Head*

5.2.1. Equations of the Motion in the Two-Dimensional Plane

After the debris flow rushes into the main river, the front of the debris blow has a certain morphology. The end of the debris flow can be considered to form an interface with the water flow since the mixing of the debris flow by the water flow is ignored. Hence, the debris flow and the water flow can be analyzed as two separate fluids in motion and coupled according to their interaction characteristics.

The motion pattern of the debris flow head is illustrated in Figure 8, based on the model tests and field studies. Disregarding the material exchange between the debris flow and the riverbed, the motion equation of the mixed flow can be obtained as follows:

$$\frac{\partial}{\partial t}(\rho_d u) + \frac{\partial}{\partial y}\left(\rho_d u^2\right) + \frac{\partial}{\partial z}(\rho_d uw) = \rho_d g \sin\theta - \frac{\partial p}{\partial y} - \frac{\partial}{\partial z}\tau_{zx} \tag{15}$$

$$\frac{\partial}{\partial t}(\rho_d w) + \frac{\partial}{\partial y}(\rho_d uw) + \frac{\partial}{\partial z}\left(\rho_d w^2\right) = -\rho_d g \cos\theta - \frac{\partial p}{\partial z} - \frac{\partial}{\partial y}\tau_{zx} \tag{16}$$

where $p$ and $\tau_{zx}$ are the static pressure and shear stress at the interface between the debris flow and the water flow, respectively.

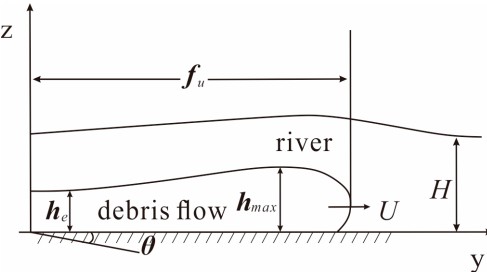

**Figure 8.** Illustration of the motion of the debris flow head.

5.2.2. Basic Equation of the Debris Flow Head

Substituting the momentum equations and mass equations into the model shown in Figure 8, and integrating the mass conservation equations for solids and mixed flows from the riverbed to the interface between the debris flow and water flow, while the volume concentration of solid particles can be considered to be close to zero as it is small at the interface, gives:

$$\frac{\partial}{\partial t}\int_0^h C_V dz + \frac{\partial}{\partial x}\int_0^h u C_V dz = -w_s C_{Vb} \tag{17}$$

$$\frac{\partial}{\partial t}\int_0^h \rho_d dz + \frac{\partial}{\partial x}\int_0^h \rho_d u dz = \rho_d w_e - \rho_s w_s C_{Vb} \tag{18}$$

where $u_p, w_p$ are the y- and z-directional flow rates at the debris flow and water flow interfaces, respectively; $w_b$ is the z-directional flow rate at the riverbed; and $C_{Vb}$ and $C_{Vp}$ are the volume concentrations of debris flow materials at the water flow interface and at the riverbed bottom, respectively.

For Equations (17) and (18), only the debris flow head is integrated in the y-direction, and assuming that the debris flow body does not change over time, $\frac{\partial y_f}{\partial t} = u_f$, $\frac{\partial y_e}{\partial t} = u_e = v_e$ ($v_e$ is the flow rate of the debris flow body), Equations (17) and (18) can be simplified as

$$\frac{\partial}{\partial t}\int_{y_e}^{y_f}\int_0^h C_V dz dy = -\int_{y_e}^{y_f} w_s C_{Vb} dy \tag{19}$$

$$\frac{\partial}{\partial t}\int_{y_e}^{y_f}\int_0^h \rho_d dz dy = \int_{y_e}^{y_f} \rho_d w_e dy - \int_{y_e}^{y_f} \rho_\sigma w_s C_{Vb} dy \tag{20}$$

$$\frac{\partial}{\partial t}\int_{y_e}^{y_f}\int_0^h u\rho_d dzdy = \rho gR\sin\theta\int_{x_e}^{x_f}\int_0^h C_V dzdy - \tfrac{1}{2}\rho gR\cos\theta C_V \overline{h}^2$$
$$-\int_0^{h_f} p'dz + \int_0^{h_e} p'dz - \int_{x_e}^{x_f} z_p dy + \int_{x_e}^{x_f} z_b dy \tag{21}$$

*5.3. Travel Distances of the Debris Flow Head*

Since the measured data of submerged-type debris flows are limited, accurate characterization analysis on the debris flow head is difficult. Liu [27] studied the characteristics of submerged flows using flume tests and proposed a simplified method for the basic equations of debris flow heads based on the test results.

By ignoring the head speed, Equations (19) and (20) can be simplified as

$$\frac{d}{dt}\left(p_0\overline{A}\right) = 0 \tag{22}$$

$$\frac{d}{dt}\left(c\overline{A}\right) = 0 \tag{23}$$

Simplifying the pressure and friction terms in Equation (21), a simplified equation can be obtained:

$$\frac{d}{dt}\left(\rho_d U\overline{A}\right) = \rho gR\sin\theta C\overline{A} + \rho gR\cos\theta C_v\overline{h}^2/2$$
$$-\rho C_D U_f^2 A_z - f_p \rho U^2 P_{\mathrm{P}}/2 - f_b \rho U^2 P_{\mathrm{b}}/2 \tag{24}$$

where $f_b, f_p$ are the coefficients of friction at the intersection and riverbed, respectively; and $P_P, P_b$ are the wetted lengths at the intersection and the riverbed, respectively.

Assuming that the average head height is constant and considering that there is a functional relationship between the head geometry and head height yields

$$L_f = \xi_f h_{\max}, L_e = \xi_e h_{\max}$$
$$\overline{A} = \xi_{\overline{A}} h_{\max}^2, A_z = \xi_{A_z} h_{\max}^2$$
$$P_p = \xi_p h_{\max}, P_b = \xi_b h_{\max}$$

After substitution of these relations into Equations (20)–(24), these equations can be simplified as

$$\frac{d}{dt}\left(\rho_d h_{\max}^2\right) = 0, \frac{d}{dt}\left(C h_{\max}^2\right) = 0 \tag{25}$$

$$\xi_{\overline{A}}\frac{d}{dt}\left(\rho_d U h_{\max}^2\right) = \rho gR\sin\theta C\xi_{\overline{A}} h_{\max}^2 + \tfrac{1}{2}\rho gR\cos\theta C_v h_{\max}^2$$
$$-\rho C_D\left(U + \xi_f\frac{dh_{\max}}{dt}\right)^2\xi_{A_z} h_{\max}^2 - \tfrac{1}{2}f_p\rho U^2\xi_p h_{\max} - \tfrac{1}{2}f_b\rho U^2\xi_b h_{\max} \tag{26}$$

The changes in density and volume concentration of the mixed flow are not considered when the velocity of the river water is low, and the analytical solution to Equations (25) and (26) is

$$U = \frac{\sqrt{a}\left(\sqrt{a} + \sqrt{c}U_0\right)e^{2\sqrt{ac}t} - \left(\sqrt{a} - \sqrt{c}U_0\right)}{\sqrt{c}\left(\sqrt{a} - \sqrt{c}U_0\right) + \left(\sqrt{a} + \sqrt{c}U_0\right)e^{2\sqrt{ac}t}} \tag{27}$$

where

$$a = \left[gR\sin\theta C_{V0}\xi_{\overline{A}} h_0^2 + \tfrac{1}{2}gR\cos\theta C_{V0} h_e^2\right]\big/\left[\xi_{\overline{A}}\left(1 + RC_{V0} h_0^2\right)\right]$$
$$c = \left[C_D\xi_{\overline{A}} h_0^2 + \tfrac{1}{2}f_p\xi_p h_0 + \tfrac{1}{2}f_b\xi_b h_0\right]\big/\left[\xi_{\overline{A}}\left(1 + RC_{V0} h_0^2\right)\right]$$

The velocity sequence $\{U_i\}$ of the debris flow is calculated at interval $\Delta T$. By using Equation (27), the maximum travel distance of the debris flow along the river after the debris flow entered the river can be calculated as

$$l_{\max} = \sum_{i=1}^{n} U_i \Delta T_i \sin \beta \qquad (28)$$

## 6. Identification Method of River Blocking

The motion equations and maximum travel distance for dam-type and submerged-type debris flows are given in this study. A comprehensive method to determine the degree of river blocking by a debris flow is proposed herein, taking into account two factors: the length of the debris flow deposit and the amount of material.

(1) Maximum blocking degree

According to Equation (28), the maximum travel distance $l_{max}$ can be calculated, and then, the maximum blocking degree S can be defined as

$$S = \frac{l_{\max}}{L_{\text{width of main river}}} \qquad (29)$$

where $S \in [0, 1]$.

Although this index can reflect the degree of river blocking by debris flow, it is still insufficient to represent the actual river blocking capacity because only the maximum travel distance is discussed, and the water interception capacity of the section is not considered. Therefore, the material conditions needed for river blocking should be considered in order to establish a more reasonable criterion.

(2) Source volume

Zhou et al. [30] proposed a formula for calculating the total amount of debris flow deposition in the main river:

$$V_b = \left( \frac{1}{2 \tan 14°} + \frac{1}{2 \tan \phi} \right) H^2 L' \qquad (30)$$

This equation is applicable when the river is completely blocked, but it is no longer applicable when a local blockage actually occurs in practice. This study improves the formula based on the model illustrated in Figure 9 by replacing the critical slope value calculated in Equation (14) with the 14° assumed in Equation (30), and assuming the minimum height of the debris flow head is $h_f$, the lateral accumulation slope of the debris flow along the river is determined as

$$\tan \psi = \frac{H - h_f}{l_f}$$

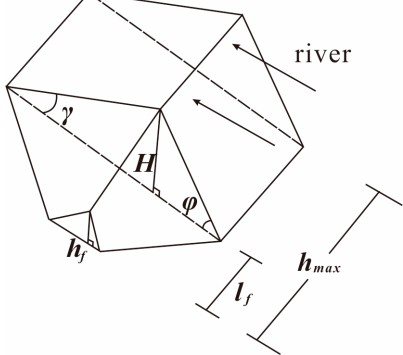

**Figure 9.** Calculation of debris flow deposits.

By establishing a local coordinate axis, the calculation formula of the height of different parts of the debris flow head can be obtained as

$$h(x) = H - x \tan \psi$$

where $x \in \left(0, l_f\right)$.

Equation (28) calculates the transport distance after the debris flow enters the river. In this study, the transport distance after the debris flow entered the river ($l_{max}$) is classified according to the correlation between the transport distance after the debris flow entered the river and the opposite bank, which is divided into three categories in total, as described below.

Equation (28) is used to calculate the travel distance of the debris flow after merging with the river. In this study, the travel distance ($l_{max}$) of debris flows after merging with rivers is classified according to the relative relationship between the debris flow and the opposite bank, which is divided into three categories as follows:

First, $l_{max} \in \left(l_f, L'\right)$, if the debris flow has not reached the opposite bank, the volume of its locally blocked dam can be calculated by the following equation:

$$
\begin{aligned}
V_{b1} &= \left(\frac{1}{2\tan\gamma} + \frac{1}{2\tan\phi}\right) H^2 \left(l_{max} - l_f\right) + \left(\frac{1}{2\tan\gamma} + \frac{1}{2\tan\phi}\right) \int_0^{l_f} h^2(x)dx \\
&= \left(\frac{1}{2\tan\gamma} + \frac{1}{2\tan\phi}\right) \left(H^2 \left(2l_{max} - l_f\right) - l_f^2 H \tan\psi + \frac{l_f^3}{3}\tan^2\psi\right)
\end{aligned}
\tag{31}
$$

Second, $l_{max} \in \left[L', L' + l_f\right)$, if the front of the debris flow head reaches the opposite bank but has not yet formed a complete river-blocking dam, the volume of the dam can be calculated by the following formula:

$$
\begin{aligned}
V_{b2} &= \left(\frac{1}{2\tan\gamma} + \frac{1}{2\tan\phi}\right) H^2 \left(l_{max} - l_f\right) + \left(\frac{1}{2\tan\gamma} + \frac{1}{2\tan\phi}\right) \int_0^{L'-l_{max}+l_f} h^2(x)dx \\
&= \left(\frac{1}{2\tan\gamma} + \frac{1}{2\tan\phi}\right) \left(H^2 L' - \left(L' - l_{max} + l_f\right)^2 H \tan\psi + \frac{\left(L'-l_{max}+l_f\right)^3}{3}\tan^2\psi\right)
\end{aligned}
\tag{32}
$$

Third, if $l_{max} \geq L' + l_f$, the debris flow can be considered to be blocking the river, and the volume of the dam is the same as in Equation (31), replacing only the critical slope:

$$V_{b3} = \left(\frac{1}{2\tan\gamma} + \frac{1}{2\tan\phi}\right) H^2 L' \tag{33}$$

As the debris flow is scoured and mixed by the water flow as it enters the main river, some of the solid materials in the debris flow are carried away by the water flow. Therefore, the total amount of actual materials required to form a debris flow dam needs to be greater than the amount of debris flow that is deposited in the main river. The definition of the loss coefficient is therefore defined as follows:

$$r = \frac{V_w}{V_w + V_b} \tag{34}$$

where $V_w$ is the volume of solids washed away by the river and $V_b$ is the volume of debris flow solids forming the dam.

From the above equation, it can be seen that the determination of the loss coefficient is of great significance to the analysis of the material amount required for debris flow blockage. Based on the field model tests in the Dadu River basin, a correlation between measured loss coefficients and river blocking coefficients is obtained, as shown in Figure 10.

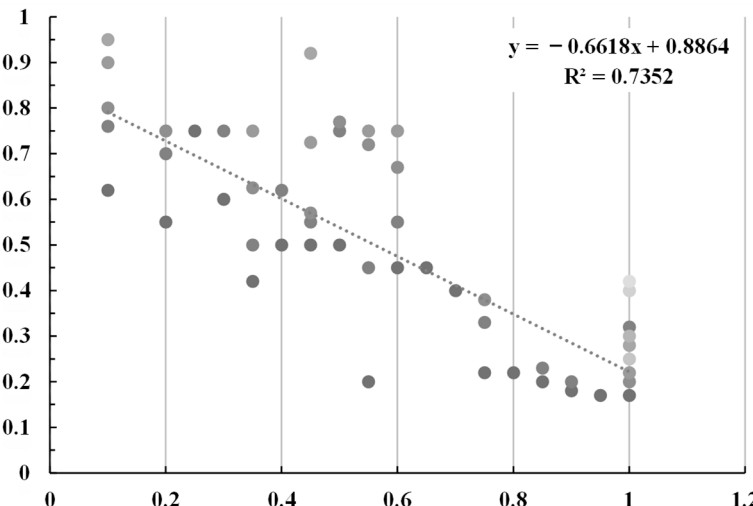

**Figure 10.** Correlation between the loss coefficient and the blocking degree based on the field model tests in the Dadu River basin.

As presented in Figure 10, there is a significant negative correlation between the loss coefficient and the blocking degree. The loss coefficient is difficult to obtain in practice, and the maximum blocking degree coefficient obtained from Equation (29) can be used to estimate the loss coefficient by the correlation presented in Figure 10.

The amount of materials required to form the dam can be calculated using Equations (31)–(33) and combined with the loss coefficient to give the total amount of material:

$$V_n = \frac{V_{bn}}{1 - r} \tag{35}$$

where n is the classification of debris flow travel distance intervals; $V_{b1}, V_{b2}, V_{b3}$ are the amount of debris flow materials that are deposited in the main river under conditions where the head of the debris flow does not reach the opposite bank, reaches the opposite bank but does not blocked, and reaches the opposite bank and is blocked, respectively; and $V_1$, $V_2$, and $V_3$ are the total amount of source materials required to form a debris flow dam under conditions where the debris flow head does not reach the opposite bank, reaches the opposite bank but does not block the river, and reaches the opposite bank and blocks the river, respectively.

On the basis of the maximum blocking degree coefficient and the amount of material sources required, the identification criterion of the river blocking by debris flow is proposed as follows:

When $S = 1$, $V > V_3$, completely blocked;
When $S = 1$, $V > V_2$, mostly blocked;
When $S < 1$, $V > V_1$, half blocked.

## 7. Discussion

Currently, the assessment of debris flow hazard intensity and extent primarily rely on empirical formulas, statistical analogies, and numerical analyses based on dynamic processes. While empirical formulas and statistical analogies can estimate the velocity and travel distance of debris flows, they often exhibit significant errors. For the movement distance of a single debris flow event, the error in empirical formulas may exceed 50% (Guo et al., 2014). Thus, we established the method for the identification of river blocking by debris flow through the modeling test carried out in the Dadu River. This method has an advantage over other methods in that it is easier to compute, provides a quicker assessment of the debris flow, and expedites the provision of emergency relief. Nevertheless, the loss coefficient of this method is calculated through model experiments, and it is necessary to supplement the debris flow data in the Dadu River area to improve its accuracy.

There are many softwares, such as Mass flow2.5, flow 3D 2022R1 and ANSYS CFX2021, which can be used to simulate debris flow blocking rivers, among which the Massflow software demonstrates high computational efficiency, supports secondary development, and has been applied in the numerical analysis of real geological hazards [31]. Subsequently, we will employ the comprehensive analysis method via the software and examine the case of debris flow in the Dadu River in order to verify its precision and suitability.

## 8. Conclusions

In this study, an identification method for river blocking by debris flows in the middle reaches of the Dadu River is proposed, and the following conclusions can be drawn:

1. The calculation formulas for the maximum travel distance of the two kinds of debris flows entering the river are obtained through theoretical derivation.
2. The formulas for calculating the length and volume of debris flow accumulation are derived, and the relationship between the debris flow loss coefficient and river blocking degree in the middle part of the Dadu River is analyzed.
3. Based on the relationship between the maximum blocking degree coefficient and the amount of material sources needed, the identification criterion of river blocking by debris flow is proposed.
4. This identification method can swiftly identify a complete dam blockage in a river, but more examples are needed to adjust its loss coefficient.

**Author Contributions:** Conceptualization, G.F. and Z.S.; methodology, Z.S. and Y.C.; software, Y.C.; validation, G.F.; formal analysis and investigation, G.F. and Z.S.; resources, G.F.; writing—original draft preparation, Y.C. and D.L.; writing—review and editing, G.F. and D.L.; visualization, Y.C.; supervision, Z.S. All authors have read and agreed to the published version of the manuscript.

**Funding:** This research was funded by the National Natural Science Foundation of China (U20A20111).

**Data Availability Statement:** Data are contained within the article.

**Conflicts of Interest:** The authors declare no conflict of interest.

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
