# Peer review of "Identification Method of River Blocking by Debris Flow in the Middle Reaches of the Dadu River, Southwest of China"

_water, doi:10.3390/w15244301_

Round 1

Reviewer 1 Report

Comments and Suggestions for Authors

In this paper, the authors aim to propose an identification method for river blocking by debris flow in Dadu River area. The debris flows were divided into dam-type and submerged debris flows based on previous large-scale model tests. Based on theoretical derivation, the authors present the formulas for calculating the maximum travel distance of the two kinds of debris flows entering the river. However, some of its details need to be refined, such as the definition of S, V1, V2, V3 in lines 22-23 are needed to explain, if not, the reader cannot carefully understand the meaning. The longitude and latitude of the study area should be added in Figure 1. Moreover, some figures and the language in this manuscript should be checked. Taking these factors into consideration, I suggest this manuscript should be returned to the author for major revisions.

1. Comment on Abstract: The first sentence in abstract is too farfetched. Of course, establishing an identification method of river blocking by debris flow in Dadu River is important. You should explain the research gap during this process, such as the most difficult point to build this method, and explain how the authors try to solve it, this can make the reader more convinced.

2. Comment on Abstract: Are the formulas for calculating the length and volume of debris flow examined by measured debris flow process?

3. Comment on Abstract: Please avoid presenting the undefined numbers or items in abstract, such as the S, V, V3, V1, since the readers cannot understand the means of those items or numbers. If not, please try to define those numbers or items in abstract.

4. Comment on introduction: Lines 29-30, debris flows can also be triggered by snow melting.

5. Comment on Introduction: There are many debris flow occurred in mountainous area in China, such as in the eastern Pamirs. Maybe the author can introduce the debris flow distribution pattern in China at the start of the introduction. The authors can find for information form published literature (https://doi.org/10.1016/j.catena.2023.106911; https://doi.org/10.1007/s10346-023-02030-w2).

6. Comment on Study area: Please add a Figure to present the location of your study area, since the reader who is not familiar with this study area cannot know the exact position of Dadu River. You also can hint other elements of the study area in this figure, such as the elevation, geologic data, and so on. I believe you can find great examples of how to design a great figure to introduce your study area (https://doi.org/10.1016/j.catena.2021.105830; https://doi.org/10.1016/j.catena.2021.105229; https://doi.org/10.1029/2022JF007047).

7. Comment on Figure 1. Please add the longitude and latitude of your study area to Figure 1.

8. Comment on Figure 2. This sketch figure is too simple, please modify this Figure, such as you can add the necessary annotation to explain the meaning of the sign in this Figure.

9. Comment on conclusions. The conclusion of number 1 is not inappropriate here, since this is just a classification standard to divide the type of debris flow, it should not be present in conclusion part.

10. Comment on conclusions: The conclusions part should modify; the main finding of your study should be present in several single sentences, not three or four long paragraphs.

Author Response

Thank you very much for taking the time to review this manuscript. We have revised the manuscript according to your comments. Please refer to the attachment for the revised manuscript.

Reviewer 2 Report

Comments and Suggestions for Authors

Dear Authors!

It is a very interesting work of You which contains a well-based theorethical approach for the analyses of debris-flow. I think later (in a different research work) maybe You should verify Your thesis with other methods in a practical way.

There are some small comments in the attached documents.

Author Response

Thank you very much for taking the time to review this manuscript. We have made changes to the manuscript based on your comments, and please see the attached document for the revised manuscript.

Round 2

Reviewer 1 Report

Comments and Suggestions for Authors

The manuscript has been improved significantly according my comments. I believe it should be accepted now.

Author Response

(The authors gave the same response as above.)
